# Interaction-based Mendelian randomization with measured and unmeasured gene-by-covariate interactions

**Wes Spiller** [1¤]*, **Fernando Pires Hartwig** [2], **Eleanor Sanderson** [1], **George Davey Smith** [1], **Jack Bowden** [3]

**1** Population Health Sciences, University of Bristol, Bristol, United Kingdom, **2** Postgraduate Program in Epidemiology, Federal University of Pelotas, Pelotas, Brazil, **3** University of Exeter Medical School, Exeter, United Kingdom

¤ Current address: MRC Integrative Epidemiology Unit, University of Bristol, Bristol, United Kingdom
* wes.spiller@bristol.ac.uk

**Data Availability Statement:** Code for performing simulations presented in the paper can be found at https://github.com/WSpiller/GxE_Simulation. Individual level data for applied analyses used data

## Abstract

Studies leveraging gene-environment (GxE) interactions within Mendelian randomization (MR) analyses have prompted the emergence of two similar methodologies: MR-GxE and MR-GENIUS. Such methods are attractive in allowing for pleiotropic bias to be corrected when using individual instruments. Specifically, MR-GxE requires an interaction to be explicitly identified, while MR-GENIUS does not. We critically examine the assumptions of MR-GxE and MR-GENIUS in the absence of a pre-defined covariate, and propose sensitivity analyses to evaluate their performance. Finally, we explore the effect of body mass index (BMI) upon systolic blood pressure (SBP) using data from the UK Biobank, finding evidence of a positive effect of BMI on SBP. We find both approaches share similar assumptions, though differences between the approaches lend themselves to differing research settings. Where a suitable gene-by-covariate interaction is observed MR-GxE can produce unbiased causal effect estimates. MR-GENIUS can circumvent the need to identify interactions, but as a consequence relies on either the MR-GxE assumptions holding globally, or additional information with respect to the distribution of pleiotropic effects in the absence of an explicitly defined interaction covariate.

## Introduction

Mendelian randomization (MR) is an epidemiological approach applied to observational data, wherein genetic variants are used as instrumental variables (IVs) to estimate the effect of a modifiable exposure on a downstream outcome [1]. MR encompasses a wide range of statistical methods, and typically relies upon three assumptions to test for causality. A suitable genetic IV is strongly associated with the exposure of interest (IV1), independent of confounders of the exposure and outcome as well as confounders of the genetic IV and outcome (IV2), and independent of the outcome when conditioning on the exposure (IV3) [1, 2].

from the UK Biobank (Application number 8786).
UK Biobank data are available from the UK Biobank
team to researchers who meet the criteria for
access to confidential data. Further information can
be found at https://www.ukbiobank.ac.uk/enable-
your-research/apply-for-access.

**Funding:** Wes Spiller is supported by a Wellcome
Trust studentship (108902/B/15/Z).

**Competing interests:** The authors have declared
that no competing interests exist.

Violation of assumptions IV2–3 can introduce bias into MR effect estimates, and as a consequence methods for identifying and correcting for such bias have formed a central theme within the MR methods literature [2–4]. In many cases, such methods focus upon correcting for bias resulting from associations between a genetic IV and an outcome which are unrelated to the exposure of interest, defined as horizontal pleiotropic pathways [5]. Pleiotropy robust methods frequently use heterogeneity in causal effect estimates across multiple genetic IVs as an indicator of horizontal pleiotropy, though such approaches are less feasible as the number of available genetic IVs decreases [6, 7].

One solution to the problem of limited available genetic IVs is to leverage variation in instrument strength across one or more covariates within a target population, representing a gene-by-covariate interaction [8–10]. Intuitively, were it possible to identify a population subgroup for which a genetic IV and exposure are independent (i.e., a 'no-relevance group'), it follows that, in the absence of horizontal pleiotropy, the genetic IV and outcome should also be independent. A non-zero instrument-outcome association for such a group would therefore be indicative of pleiotropic bias [8, 11, 12]. It is, however, rare that no-relevance groups of sufficient size are observed in practice.

MR approaches utilising gene-by-covariate interactions, here referred to as interaction-MR, overcome this limitation by using statistical assumptions to extrapolate back to a hypothetical no-relevance group. Two such methods are MR using Gene-by-Environment interactions (MR-GxE) and MR G-Estimation under No Interaction with Unmeasured Selection (MR-GENIUS) [8, 13]. MR-GxE uses an explicitly defined gene-by-covariate interaction to estimate causal effects, and has previously been framed within a summary-level data context [8]. In contrast, MR-GENIUS accommodates both observed and unobserved interactions, provided they induce a dependence between the genetic IV and exposure variance [13]. MR-GENIUS has the advantage of circumventing the need to explicitly identify gene-by-covariate interactions, though the relative strengths and limitations of the approach compared to MR-GxE have previously been unclear.

In this paper we outline the implementation of MR-GxE in the individual level data setting, and critically evaluate the performance of MR-GxE and MR-GENIUS. Specifically, we focus upon the application of MR-GENIUS in the absence of a pre-defined interaction covariate, which is not possible using MR-GxE. Through simulation we demonstrate how both approaches share similar underlying assumptions, and highlight how implicitly leveraging all potential gene-by-covariate interactions using MR-GENIUS can imply more stringent assumptions with respect to the distribution of pleiotropic effects. Throughout we also propose sensitivity analyses to test the assumptions of the MR-GxE. Finally, we conduct applied analyses using MR-GxE and MR-GENIUS to estimate the effect of body mass index (BMI) on systolic blood pressure (SBP). For both approaches we find evidence of a positive causal effect using data from the UK Biobank, comparing results to conventional MR and observational methods.

## Materials and methods

### The data generating model

Interaction-MR approaches use differences in instrument strength across one or more covariates to estimate and correct horizontal pleiotropic bias [8]. For $i \in \{1, 2, \ldots, N\}$ observations, let $G_i$ denote a single genetic IV for an exposure $X_i$, and let $Y_i$ represent the outcome of interest. Further, assume there exists an unmeasured confounder $U_i$ of $X_i$ and $Y_i$, and a set of interaction covariates $Z_i \in \{Z_1, \ldots, Z_K\}$ across which the instrument-exposure association varies. In order to make our ideas concrete, we now define an underlying data generating model for a

continuous exposure and outcome, which are themselves a function of $G_i$, $Z_i$ and $U_i$.

$$Z_{ki} = \pi_{k0} + \pi_{k1} G_i + \epsilon_{Zki} \tag{1}$$

$$U_i = \theta_0 + \theta_1 G_i + \sum_{k=1}^{K} (\theta_{2k} Z_{ki} + \theta_{3k} G_i Z_{ki}) + \epsilon_{Ui} \tag{2}$$

$$X_i = \gamma_0 + \gamma_1 G_i + \sum_{k=1}^{K} (\gamma_{2k} Z_{ki} + \gamma_{3k} G_i Z_{ki}) + \gamma_4 U_i + \epsilon_{Xi} \tag{3}$$

$$Y_i = \beta_0 + \beta_1 X_i + \beta_2 G_i + \sum_{k=1}^{K} (\beta_{3k} Z_{ki} + \beta_{4k} G_i Z_{ki}) + \beta_5 U_i + \epsilon_{Yi} \tag{4}$$

In Eqs 1–4, the $\epsilon_{(.i)}$ terms represent independent error terms, and relationships with reference to a single interaction covariate $Z_{ki}$ are illustrated in Fig 1 wherein $G_i$, $Z_{ki}$, and $U_i$ are assumed independent for clarity.

## An overview of MR-GxE and MR-GENIUS

The MR-GxE and MR-GENIUS approaches rely upon one or more first-stage interactions which induce variation in the association between the genetic IV and exposure. Specifically, the MR-GxE approach requires an interaction covariate ($Z_k i$) to be explicitly observed, in contrast to MR-GENIUS which leverages variance differences for a given exposure ($X_i$) across

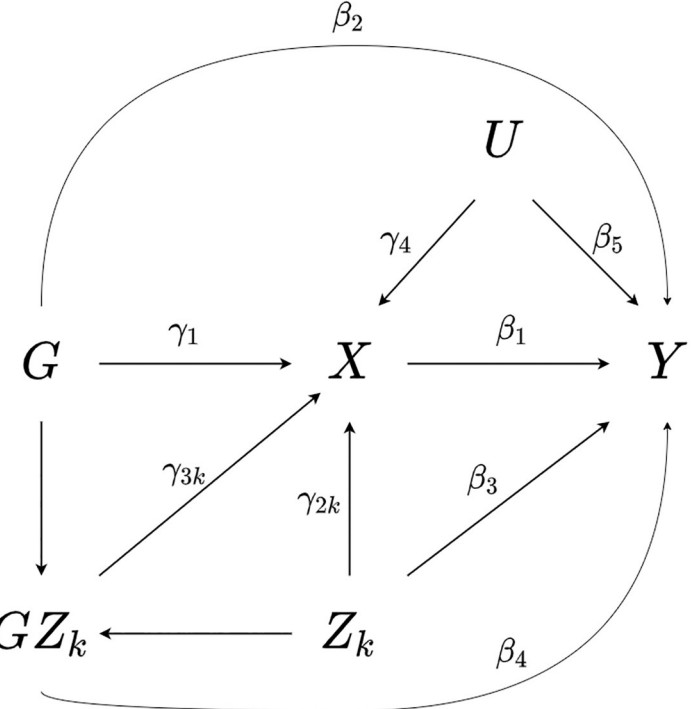

**Fig 1. Illustration of data generating model.** A directed acyclic graph showing the relationship between a genetic instrument $G$, an interaction covariate $Z_k$, exposure $X$, outcome $Y$, and one or more confounders $U$. $GZ_k$ denotes the interaction $G \times Z_k$, and $G$, $Z_k$, and $U$ are assumed independent.

subgroups of a genetic IV ($G_i$). In this paper we illustrate how both approaches are reliant upon three assumptions, summarised as assumptions GxE1–3 below. A suitable interaction ($G_i Z_{ki}$) is:

1. Strongly associated with the exposure of interest (GxE1).

2. Independent of confounders of the exposure and outcome (GxE2).

3. Not directly associated with the outcome of interest (GxE3).

MR-GxE was originally implemented using an approach analogous to MR-Egger regression in two-sample summary MR [7, 8]. Initially sets of instrument-exposure and instrument-outcome associations are obtained across strata of a pre-specified interaction covariate, after which the instrument-outcome associations are regressed upon the instrument-exposure associations including an intercept [8]. While in principle the approach can be performed using publicly available data from genome-wide association studies (GWAS), the summary level MR-GxE approach has two notable limitations. First, summary MR-GxE does not readily provide a means of evaluating interaction strength, relying on observed heterogeneity across gene-exposure associations across interaction covariate strata [8]. Second, ambiguities surrounding the optimal number of interaction covariate strata can have a substantial impact of effect estimates [8]. To address these issues, we propose an individual-level form of MR-GxE within a two-stage least squares (TSLS) framework.

Individual level MR-GxE is implemented by using a gene-by-covariate interaction as an instrument within a TSLS regression model. In the first-stage model (Eq 5), the exposure is regressed upon the genetic IV and observed interaction covariate including an interaction term ($\gamma_3$). The second-stage model (Eq 5) then regresses the outcome upon the genetic IV, interaction covariate, and fitted values for the exposure ($\hat{X}_i$) obtained using the first-stage model.

$$\hat{X}_i = \gamma_0 + \gamma_1 G_i + \gamma_2 Z_{ki} + \gamma_3 G_i Z_{ki} \tag{5}$$

$$Y_i = \beta_0 + \beta_1 \hat{X}_i + \beta_2 G_i + \beta_3 Z_{ki} + \epsilon_{Yi} \tag{6}$$

This returns a causal effect estimate ($\hat{\beta}_1$), as well as a horizontal pleiotropic effect estimate as the coefficient of the genetic IV ($\hat{\beta}_2$) in the second-stage model. To define the MR-GxE estimand, a reduced form model for $Y_i$ given $G_i$ and $Z_{ki}$ incorporating Eqs 5 and 6 can initially be written as

$$Y_i = \alpha_0 + \alpha_1 G_i + \alpha_2 Z_{ki} + \alpha_3 G_i Z_{ki} + \epsilon_i \tag{7}$$

Using Eqs 5 and 6 the MR-GxE estimand is then defined as

$$\beta_1 = \frac{cov(Y, GZ_k) - \alpha_1 cov(G, GZ_k) - \alpha_2 cov(Z_k, GZ_k)}{cov(X, GZ_k) - \gamma_1 cov(G, GZ_k) - \gamma_2 cov(Z_k, GZ_k)} \tag{8}$$

Note that a $G_i \times Z_{ki}$ term is omitted from the second-stage model given in Eq 6 due to its role as an instrument, whilst the inclusion of $G_i$ allows for estimation of a horizontal pleiotropic effect on the outcome, denoted by $\beta_2$.

The MR-GENIUS approach is an adapted form of Robins' G-estimation which is robust to additive confounding and pleiotropic bias [13–15]. This essentially involves leveraging differences in the variance of a given exposure $X_i$ across subgroups of a genetic instrument $G_i$, which are likely the consequence of one or more gene-by-covariate interactions. In the case of

a binary instrument and exposure, and using notation from Eqs 1–4, the MR-GENIUS estimator can be written as:

$$\hat{\beta}_1 = \frac{\mathbb{P}_n[\{G_i - \mathbb{P}_n(G)\}\{X_i - \hat{E}(X_i|G_i)\}Y_i]}{\mathbb{P}_n[\{G_i - \mathbb{P}_n(G_i)\}\{X - \hat{E}(X_i|G_i)\}X_i]} \tag{9}$$

where $\mathbb{P}_n = n^{-1} \sum_{i=1}^n [\bullet]_i$ and $\hat{E}(X_i|G_i = g) = \mathbb{P}_n[X_i 1(G_i = g)]/\mathbb{P}_n[1(G_i = g)]$ [13].

MR-GENIUS is implemented by first regressing $X_i$ upon $G_i$ and obtaining a set of residuals $\hat{\epsilon}_{Xi}$. These residuals are then used to create an instrument $(G_i - \bar{G})\hat{\epsilon}_{Xi}$ which is incorporated within a TSLS model as a single instrument for $X_i$ [13]. Estimates of $\hat{\beta}_1$ remain unbiased, provided the instrument $G_i$ is associated with the exposure of interest, the effect does not change across values of the unmeasured confounders, and the MR-GENIUS model is identified such that the change in variance across levels of the instrument is non-zero [13].

In the binary exposure case, the MR-GENIUS model is identified when $cov(G_i, var(X_i|G_i)) \neq 0$, and for a continuous exposure when the residual error $\epsilon_X$ is heteroskedastic, that is, not constant across levels of $G_i$ [13]. This can be evaluated using a Breusch-Pagan test for heteroskedasticity, and these conditions also restrict the degree of joint effect modifiers of both $X_i$ and $Y_i$ [13, 16].

Importantly, it should be noted that the interaction covariate need not be explicitly identified using MR-GENIUS, illustrated by the absence of $Z_{ki}$ in Eq 9. However, identification of the MR-GENIUS model implicitly relies upon the presence of one or more gene-by-covariate interactions to induce the desired dependence between $G_i$ and $var(X|G)$. In the absence of a predefined interaction covariate, MR-GENIUS estimates the total effect of $X$ upon $Y$, without adjusting for the interaction covariate $Z_k$. This contrasts with MR-GxE, which estimates the direct effect of $X$ upon $Y$ adjusting for the interaction covariate in the second stage model.

## GxE1: Interaction strength

The MR-GxE estimator can be viewed as an extension of the Wald ratio, including an adjustment for the direct effects of $G_i$ and $Z_{ki}$. Thus, in the special case where $G_i$ and $Z_{ki}$ are marginally independent of the exposure and outcome (but their interaction via a single covariate $Z_{ki}$ is not), the MR-GxE estimator simplifies to:

$$\frac{cov(Y, GZ_k)}{cov(X, GZ_k)} \tag{10}$$

From Eq 10 MR-GxE is clearly reliant upon a strong first-stage interaction, such that $\gamma_{k3} \neq 0$ in order to make the denominator non-zero (GxE1). When individual-level data are available, the first-stage F-statistic for the gene-by-covariate interaction can be used to quantify instrument strength, though several aspects of this approach warrant consideration. First, when using a single interaction the F-statistic cannot be related to the magnitude of relative bias towards the observational estimate in a one-sample setting and null in a two-sample setting. This is because such a relationship between instrument strength and the direction of bias only holds when multiple instruments, in this case interactions, are used. Therefore, while an F-statistic of 10 may satisfy the standard threshold for sufficient instrument strength, it would not be possible to relate this to a 10% relative bias towards the observational estimate obtained by regressing the outcome on the exposure without incorporating additional interaction covariates. Second, interaction strength does not mitigate bias from violations of assumptions GxE2–3, just as is the case in conventional MR analyses. Finally, where possible candidate interactions should be identified in separate samples to avoid issues related to Winner's curse,

where instruments, in this case interactions, are selected using spurious associations which may be sample population specific [17].

The reliance of MR-GxE upon explicitly defined interactions also invites two potential interaction-specific issues: scale dependency and non-linear interactions. First, as interactions are scale dependent it is possible that applying transformations can create spurious associations [18]. Such spurious associations can exist as an artefact of the data, and consequently estimates leveraging such information are unlikely to be reliable. Gene-by-covariate interactions may also be non-linear, which could potentially be considered by fitting more flexible models (e.g., fractional polynomial models, which include varying exponents with respect to $G_i Z_{ki}$) to allow for non-linear interactions to be identified. It is, however, important to take care to avoid issues of over-fitting [19].

As MR-GENIUS does not require gene-by-covariate interactions to be identified, testing for identification is performed globally by evaluating heteroskedasticity with respect to the residuals $\epsilon_{Xi}$. Specifically, MR-GENIUS relies upon the residual error in a regression of the exposure upon the genetic IV to be heteroskedastic, evaluated using a Breusch-Pagan test for heteroskedasticity [13].

As a means of identifying candidate gene-by-covariate interactions for MR-GxE we propose using the first-stage F-statistic for the interaction term in the first stage, in a similar fashion to utilising GWASs to identify genetic variants associated with a phenotype of interest. Interactions of sufficient strength can be identified by fitting the first-stage MR-GxE model for each candidate interaction covariate $Z_{ki}$ and calculating the F-statistic with respect to $G_i Z_{ki}$ (see Eq 5) [10]. Applying a Bonferroni multiple testing correction, and plotting the $-\log_{10}(p - value)$ for the F-statistic then allows for instrument strength to be effectively visualised using a scatter plot, following a similar intuition to the use of Manhattan plots in the presentation of results from GWAS [20]. Note that as it is often the case that multiple independent genetic variants are associated, it is often appropriate to use a polygenic risk score as an instrument to maximise instrument strength.

## GxE2: Interaction exogeneity

In previous work we show how assumption GxE2 is potentially violated when certain confounding structures exist, specifically, where $G_i$ and $Z_{ki}$ are simultaneously downstream of a confounder $U_i$ or where there is an open path between the two variables through $U_i$ [8]. To briefly recapitulate how such associations can induce bias, consider the path diagram shown in Fig 2. In this case, the interaction covariate $Z_{ki}$ is independent of $X_i$ and $Y_i$, and determined by a confounder $U_i$. Further, $U_i$ is downstream associated with the genetic instrument $G_i$.

In Fig 2 $U_i$ serves not only as a confounder of $X_i$ and $Y_i$, but also of $Z_i$ and $Y_i$. As the MR-GxE model only instruments $G_i$, it is likely estimates for the effect of $Z_i$ on $Y_i$ will exhibit bias. When $G_i$ is not independent of $U_i$, however, the resulting induced association between $Z_i$ and $Y_i$ from failing to control for $U_i$ mimics a pleiotropic association, such that a pathway from $G_i$ to $Y_i$ is created through $U_i$ and $Z_i$. Importantly, such associations do not necessarily bias estimates of the effect of $X_i$ on $Y_i$, but can inflate type-I error rates when evaluating instrument validity.

Assumption GxE2 can also be violated when a gene-by-covariate interaction is simultaneously associated with the exposure and one or more confounders of the exposure and outcome, as depicted in Fig 3.

In Fig 3 a bidirectional arrow is included to highlight that any direction of association between $GZ_k$ and $U$ can potentially introduce bias into MR-GxE estimates. Where $GZ_k$ is upstream of $U$ this can be viewed as instrument strength varying across levels of the

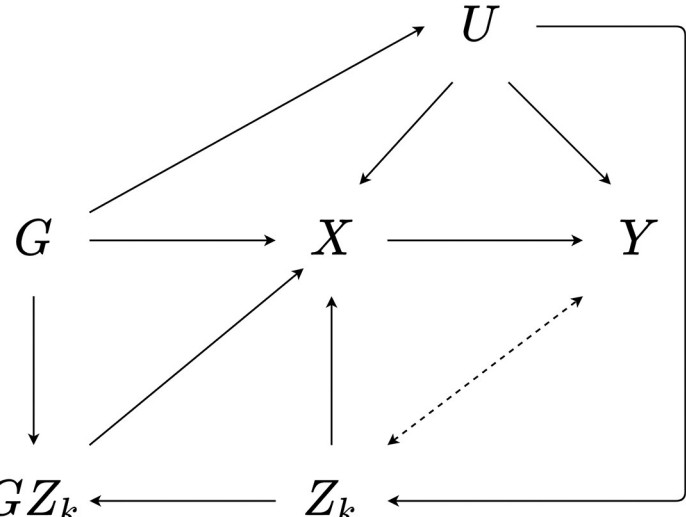

**Fig 2. An example of GxE2 through confounding.** A path diagram illustrating a case in which the instrument $G_i$ is a determinant of the interaction covariate $Z_i$ through a confounder $U_i$. The bidirectional dashed arrow from $Z_k$ to $Y$ represents an association induced due to confounding as a result of not adjusting for $U$ in the second stage MR-GxE model.

confounder, with pleiotropic effects being associated with interaction strength. This issue can be viewed as analogous to the INstrument Strength Independent of Direct Effect (InSIDE) assumption in two-sample summary MR [7]. An association from $U$ to $GZ_k$ would suggest that a three-way interaction may be present, such that interaction strength $\gamma_3 k$ varies across levels of $U$. This can bias effect estimates by inducing an association between $GZ_k$ and $Y$, violating the constant pleiotropy assumption GxE3.

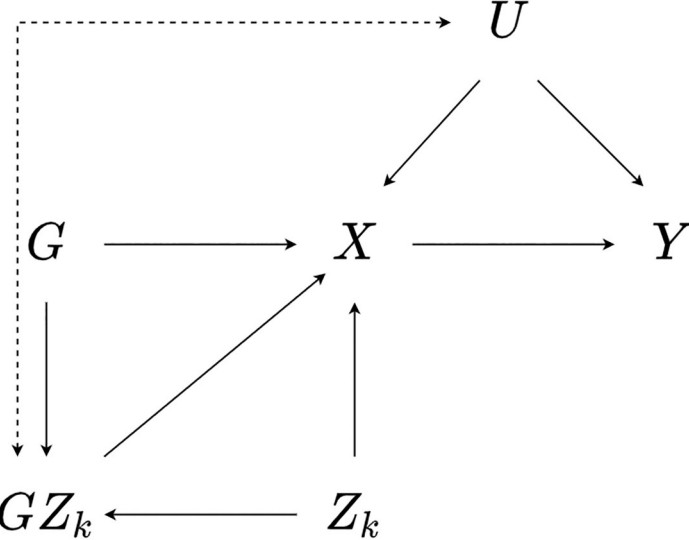

**Fig 3. Illustration of general GxE2 violation.** A directed acyclic graph showing the relationship between a genetic instrument $G$, interaction covariate $Z_k$, exposure $X$, outcome $Y$, and one or more confounders $U$. In this case, the presence of an association between a gene-by-covariate $GZ_k$ and $U$ violates assumption GxE2.

To understand how an association between $GZ_k$ and $U$ can induce bias into MR-GxE estimates, we can extend the MR-GxE estimand (Eq 8) to incorporate violation of GxE2, by including covariance terms between $U_i$ and $(Z_ki, G_iZ_ki)U_i$, such that were it possible to include $U_i$ in the TSLS model, the resulting estimate could be written as:

$$\hat{\beta}_1 = \frac{cov(Y, GZ_k) - \beta^*_{YG}cov(G, GZ_k) - \beta^*_{YZ}cov(Z, GZ_k) - \beta^*_{YU}cov(U, GZ_k)}{cov(X, GZ_k) - \beta^*_{XG}cov(G, GZ_k) - \beta^*_{XZ}cov(Z, GZ_k) - \beta^*_{XU}cov(U, GZ_k)} \tag{11}$$

where each $\beta^*_{\bullet}$ indicates a multivariable regression estimate pertaining to the second subscript variable when regressed upon the first, including the unmeasured confounder $U_i$. As it is not possible to directly measure and adjust for $U_i$, the independence $U_i$ and $G_iZ_{ki}$ is relied upon for Eq 11 to be equivalent to the MR-GxE estimator in Eq 8.

A further consideration is the introduction of collider bias when estimating fitted values $\hat{X}_i$ in the first-stage MR-GxE model. As shown in Eq 5, it is necessary to include the interaction covariate in the first-stage model. However, in cases where $G_i$ and $U_i$ are both simultaneously upstream associated with $Z_{ki}$, conditioning on $Z_{ki}$ will induce collider bias in the first-stage MR-GxE model, such that the estimate of pleiotropic effect $\hat{\beta}_2$ and subsequent adjustment will be inaccurate. This case is illustrated in Fig 4.

Relating assumption GxE2 to the MR-GENIUS approach, associations violating GxE2 would imply associations vary across values of the unmeasured confounders violating the second MR-GENIUS assumption [13]. However, this problem can be mitigated by incorporating additional interaction covariates within the MR-GENIUS model, as described in Eric Tchetgen et al., 2021 [13]. This would necessitate the inclusion of specific interaction covariates within the MR-GENIUS model, such that differences in the variance of $X$ would be evaluated across subgroups of $G_i$, conditional on one or more interaction covariates $Z_k$.

For MR-GxE we present two strategies for addressing GxE2 violation. To evaluate the possibility of collider bias in the first-stage model estimating the correlation between $G_i$ and $Z_{ki}$ could serve as an initial test for GxE2 violation. Intuitively, if $G_i$ and $Z_{ki}$ are independent, then

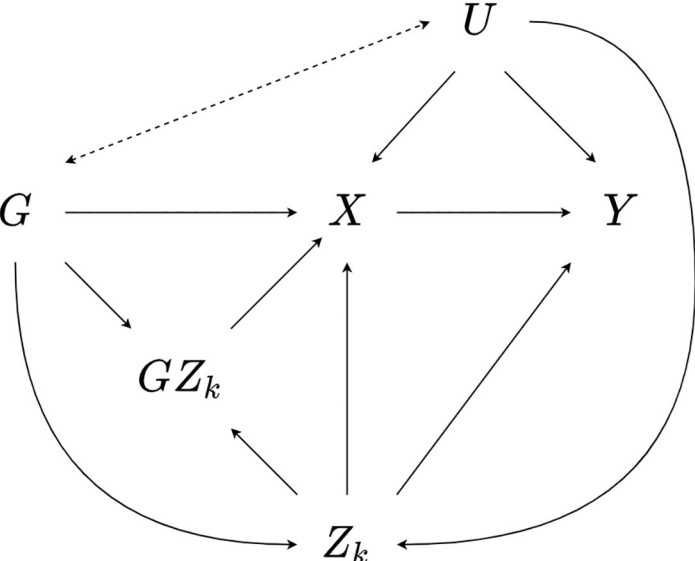

**Fig 4. Illustration of collider bias when estimating $\hat{X}$.** A diagram showing a situation in which conditioning on $Z_k$ when $G$ and $U$ are simultaneously upstream associated with $Z_k$ would induce collider bias, as shown by the dashed bidirectional arrow.

conditioning on $Z_{ki}$ would not induce an association between $G$ and $U$. However, it is important to emphasise that independence cannot necessarily be interpreted as GxE2 being satisfied. This would primarily be the case where a three-way interaction exists between the instrument $G_i$, interaction covariate $Z_{ki}$, and one or more confounders $U_i$. Rather than removing the possibility, an observed correlation between $G_i$ and $Z_{ki}$ can highlight a potential issue in the analysis which warrants further consideration.

A potentially more robust approach would be to adopt a genetic proxy variable for the interaction covariate $Z_{ki}$, as this would share the same benefits with regard to causal direction as $G_i$ with respect to environmental confounders. For example, when estimating the effect of alcohol consumption on SBP using education as an interaction covariate, adopting a polygenic risk score (PRS) for education would in principle utilise the explained variation in education excluding environmental confounders such as socio-economic status.

### GxE3: Constant pleiotropy

The third MR-GxE assumption requires pleiotropic effects of $G_i$ upon $Y_i$ to remain constant across values of $Z_{ki}$, with the gene-by-covariate interaction being independent of $Y_i$ when conditioning on $X_i$ (i.e. $\beta_4 = 0$). Where this is not the case estimates of causal effect will exhibit bias in the direction of $\beta_4$ in a similar fashion to horizontal pleiotropic bias in univariate MR analyses, equal to:

$$bias = \frac{\beta_4}{cov(X, GZ_k) - \beta_{XG}cov(G, GZ_k) - \beta_{XZ_k}cov(Z_k, GZ_k)} \tag{12}$$

By reframing MR-GxE within a TSLS framework, it is possible to apply tests of over-identification to evaluate the constant pleiotropy assumption, though this is not possible where only one instrument is available, for example, a single genetic variant. In cases where the single instrument is comprised of many instruments, such as a PRS, it is possible to examine different configurations of instruments iteratively using MR-GxE and assess heterogeneity in the set of MR-GxE estimates obtained from each iteration. These subsets of instruments are hereafter referred to as *sub-instruments*.

In this scenario, a Sargan test can be used to compare different MR-GxE estimates of the same causal parameter (the coefficient of $X_i$ in Eq 6—i.e., $\beta_1$), assuming we have more instruments than we need to consistently estimate the parameter [21]. However, it is important to note that in applying this test it is crucial for each of the sub-instruments to be sufficiently strong to overcome weak instrument bias, though practically the test can be applied where weak interactions are present if assessing the strength of individual instruments of interest.

To illustrate how over-identification tests can be applied in the context of MR-GxE, consider an extension of Eqs 5 and 6 to include an arbitrary number of sub-instruments, wherein a single instrument $\mathbf{G}_i$ is comprised of $m \in \{1, 2, \ldots, M\}$ sub-instruments. Where $G_{mi}$ denotes the $m^{th}$ sub-instrument in $\mathbf{G}_i$, we can define a corresponding data generating model as:

$$X_i = \gamma_0 + \sum_{m=1}^{M}\left(\gamma_{1m}G_{mi} + \sum_{k=1}^{K}(\gamma_{2k}Z_{ki} + \gamma_{3km}G_{mi}Z_{ki})\right) + \gamma_4 U_i + \epsilon_{Xi} \tag{13}$$

$$Y_i = \beta_0 + \beta_1 X_i + \sum_{m=1}^{M}(\beta_{2m}G_{mi}) + \sum_{k=1}^{K}(\beta_{3k}Z_{ki}) + \sum_{m=1}^{M}\sum_{k=1}^{K}(\beta_{4km}G_{mi}Z_{ki}) + \beta_5 U_i + \epsilon_{Yi} \tag{14}$$

A Sargan test can be applied by fitting multiple sub-instruments $G_{mi}$ in the same TSLS model. Alternatively, a heterogeneity test such as Cochran's Q-statistic could be used to

evaluate heterogeneity in MR-GxE effect estimates using all sets of non-overlapping sub-instruments.

## Results

### An illustration of assumptions GxE1–3 through simulation

To illustrate the importance of assumptions GxE1–3 with respect to MR-GxE and MR-GENIUS we present six simulation studies, categorised by assumption, using the data generating model presented in Eqs 1–4. Throughout, we demonstrate the utility of the sensitivity analyses proposed, and highlight the relative performance of both MR-GxE and MR-GENIUS. In each simulation gene-by-covariate first-stage effects ($\gamma_{3k}$) are generated to be positive, to avoid the possibility that the combined effects of all candidate interactions have a mean of zero. This would potentially invalidate the MR-GENIUS approach in the unlikely event that leveraged candidate interactions have effects such that $cov(G_i, var(X_i|G_i)) \approx 0$. Code for performing each simulation study and further information is available at https://github.com/WSpiller/GxE_Simulation.

**Simulation set 1: Interaction selection and strength.** As an illustration of how gene-by-covariate interactions can be identified through the evaluation of their first-stage F-statistics, we generated 1, 000 independent data sets, containing 100, 000 observations for a single instrument $G$, exposure $X$, outcome $Y$, and 100 candidate interaction covariates $Z_k$ (Simulation 1). All variables were treated as continuous, with observations of exogenous variables randomly sampled from a normal distribution with mean 0 and standard deviation 1. Endogenous variables, determined by one or more additional covariates, were generated following the population models defined in Eqs 1–4, with error terms randomly sampled from a normal distribution with mean 0 and standard deviation 1. The effect of $X$ upon $Y$ was defined as $\beta_1 = 1$. Of the 100 interaction covariates, 10 were designated to have a non-zero first-stage interaction, assigning a value for $\gamma_{3k}$ sampled from a normal distribution with mean 2 and standard deviation 2, ensuring that all coefficients for non-zero first stage interactions were greater than 1. The complete set of interaction covariates $Z_K$ were also generated so as to be independent of $G_i$, such that $\pi_{k1} = 0$. Fig 5A shows how a scatter plot can be constructed in a similar fashion to a Manhattan plot in GWAS analyses. Each value on the scatter plot represents the mean $-\log_{10}(p - value)$ value for the first-stage F-statistic corresponding to each candidate

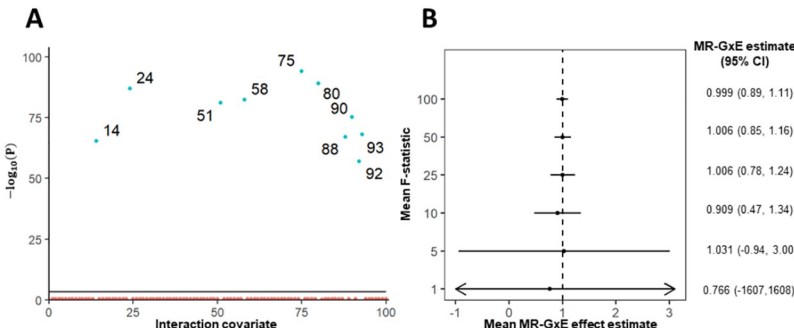

**Fig 5. Plots corresponding to simulations 1–2, identifying interactions and visualising the impact of weak instrument bias for MR-GxE.** Panel A shows a scatter plot of $-\log_{10}(p - value)$ for the mean first-stage F-statistic across the set of 100 potential interaction covariates in simulation 1. A solid horizontal line is included representing the Bonferroni correction threshold for statistical significance in panel A. Panel B shows a forest plot of mean causal effect estimates and confidence intervals under varying mean interaction strengths in simulation 2. The dotted vertical line in panel B represents the true causal effect $\beta_1 = 1$, and arrows are used to indicate confidence intervals exceeding the limits of the forest plot.

interaction across the set of 1, 000 simulated data sets. A Bonferroni multiple testing correction is shown using a solid horizontal line.

In Fig 5A the 10 defined non-zero gene-by-covariate interactions have been identified, with super-imposed numbers indicating the identity of each interaction covariate $Z_k$. The corresponding estimates for the 10 identified interactions show no evidence of apparent bias, with a mean MR-GxE estimate of 1.000 (95% CI = 0.996, 1.004) and a mean F-statistic of 3616.15 ($p-value < 0.001$). Individual mean estimates for each interaction are provided in the Supplementry material (see S1 Table). Using MR-GENIUS resulted in mean estimate of 1.000 (95% CI = 0.994, 1.006), producing an estimate comparable to MR-GxE without the need to explicitly identify an interaction covariate. Performing a Breusch-Pagan test for identification in the MR-GENIUS model yielded a mean value of 1041.74 ($p-value < 0.001$), suggesting MR-GENIUS estimates are sufficiently strong so as to overcome weak instrument bias.

To further investigate the impact of weak instrument bias using MR-GxE we perform additional simulations, evaluating the performance of MR-GxE using a single non-zero interaction covariate of varying strength (simulation 2). Specifically, first-stage F-statistics of approximately 1, 5, 10, 25, 50, and 100 are considered, generating 1, 000 data sets for each F-statistic value and presenting mean MR-GxE effect estimates and 95% confidence intervals. In each case the genetic instrument $G$ is generated so as to satisfy assumption IV1 ($\gamma_1 = 1$), with a causal effect of $X$ on $Y$ again equal to 1 ($\beta_1 = 1$). Fig 5B shows a forest plot including the mean MR-GxE effect estimate and 95% confidence interval for each interaction covariate with a mean F-statistic as indicated on the y-axis. The precision of MR-GxE increases substantially as the mean F-statistic increases, and there does not appear to be evidence of directional bias using weak interactions.

In simulation 1 MR-GENIUS appears to perform well when many gene-by-covariate interactions are present, with the potential to outperform MR-GxE when individual stronger interactions are not observed (see S1 Table). To explore the extent to which MR-GENIUS is reliant on a global non-zero mean first-stage interaction, an additional simulation is conducted varying the proportion of non-zero interactions present in the data (simulation 3). A total of $K = 100$ interactions were generated, such that the number of non-zero interactions represent 1%, 5%, 10%, 50%, and 100% of all candidate interaction covariates. For each predefined proportion, 1,000 independent data sets were generated, using previous parameter definitions from simulation 1. MR-GxE effect estimates were obtained using a single randomly sampled non-zero interaction covariate, while MR-GENIUS estimates do not specify an observed interaction covariate. The mean MR-GxE and MR-GENIUS estimates for each proportion are presented in Table 1.

From Table 1 it appears the precision of MR-GENIUS estimates improves as the mean interaction strength across all leveraged instruments ($\bar{\gamma}_K$) increases in magnitude, indicated by the increase in mean F-statistic across the set of candidate interaction covariates. This suggests that in cases where few gene-by-covariate interactions of moderate strength are available,

**Table 1. Simulated results using differing proportions of non-zero interaction covariates (simulation 3).**

| Proportion $\gamma_{3K} \neq 0$ | Mean MR-GxE $\hat{\beta}_1$ (95% CI) | MR-GENIUS $\hat{\beta}_1$ (95% CI) | Mean F-statistic | BP-Test p-value |
|---|---|---|---|---|
| 1% | 1.000 (0.99,1.01) | 1.002 (-2.34,4.34) | 6.14 | 0.202 |
| 5% | 1.000 (0.99,1.01) | 0.999 (0.96,1.04) | 23.68 | 0.007 |
| 10% | 1.000 (0.99,1.01) | 1.000 (0.98,1.02) | 39.25 | 0.002 |
| 50% | 1.000 (0.99,1.01) | 1.000 (0.99,1.01) | 76.51 | <0.001 |
| 100% | 1.000 (0.98,1.02) | 1.000 (0.99,1.01) | 87.90 | <0.001 |

MR-GxE can furnish more precise estimates than MR-GENIUS, though MR-GENIUS can outperform MR-GxE as the number of non-zero gene-by-covariate interactions increases. It is also important to highlight that, just as was the case for MR-GxE, weak instrument strength for MR-GENIUS does not appear to induce observed directional bias.

This reliance of MR-GENIUS upon mean interaction strength across candidate interactions has two important implications. First, the precision of MR-GENIUS estimates is a function of both interaction strength and the number of candidate non-zero interactions present, and consequently it is possible for MR-GENIUS to outperform MR-GxE as the number of strong gene-by-covariate interactions increases. Second, it is possible that interactions of similar magnitude acting in opposite directions can counteract each other, such that $cov(G_i, var(X_i|G_i)) \approx 0$. This scenario is unlikely to occur beyond a simulation setting, and motivates using a data generating model where first-stage interactions are generated so as to be in the same direction, to ensure $\bar{\gamma}_K \neq 0$.

**Simulation set 2: Interaction exogeneity.** The MR-GxE approach relies upon the selected gene-by-covariate interaction being independent of all confounders $U$ of the exposure $X$ and outcome $Y$. As previously discussed, this is most likely to be the case where either the association between $G$ and $U$ varies across levels of $Z_k$, or the association between $Z_k$ and $U$ varies across levels of $G$. To demonstrate how such associations can introduce bias into both MR-GxE and MR-GENIUS effect estimates, we present a simulation with a similar structure to simulation 3, in this case varying the proportion of interactions for which $\theta_3 k \neq 0$ (simulation 4). Specifically, proportions of 0%, 1%, 5%, 10%, 50%, and 99% are considered for which $\theta_3 k = 1$, generating 1, 000 independent data sets for each proportion. For each data set, the mean MR-GENIUS and MR-GxE estimates were obtained across all interaction covariates. Additionally, a mean estimate using a randomly sampled interaction for which $\theta_3 k \neq 0$ is also presented, to illustrate how assumption GxE2 is interaction covariate specific for MR-GxE. The simulation results are given in Table 2.

From Table 2 both MR-GxE and MR-GENIUS exhibit bias in the direction of the interaction coefficient $\theta_{3k}$. MR-GENIUS appears to be more robust to GxE2 violation compared to MR-GxE, though estimates decrease in precision as the magnitude of $\theta_{3k}$ increases. When selecting a single interaction covariate for which $\theta_{3k} = 0$, MR-GxE provides an unbiased causal effect estimate, in contrast to MR-GENIUS. This can be explained by MR-GENIUS implicitly relying upon $\bar{\theta}_K \approx 0$ when an interaction covariate is not specified. As a consequence, MR-GxE appears to be capable of producing results with markedly less bias using a single GxE2 satisfying interaction, compared to MR-GENIUS where $\bar{\theta}_K = 0$.

**Simulation set 3: Constant pleiotropy.** To demonstrate the impact of GxE3 violation, as well as the utility of employing an adapted Sargan test as a sensitivity analysis, we present a two

**Table 2. Simulated results using differing proportions of non-zero gene-by-covariate interaction with respect to confounders (simulation 4).**

| Proportion $\theta_{3K} \neq 0$ | Valid MR-GxE[a] $\hat{\beta}_1$ (95% CI) | Mean MR-GxE $\hat{\beta}_1$ (95% CI) | MR-GENIUS $\hat{\beta}_1$ (95% CI) | Mean F-statistic |
|---|---|---|---|---|
| 0% | 1.000 (0.998,1.002) | 1.000 (0.990,1.010) | 1.000 (0.995,1.005) | 67.62 |
| 1% | 1.000 (0.998,1.002) | 1.003 (0.990,1.016) | 1.001 (0.995,1.008) | 67.71 |
| 5% | 1.000 (0.997,1.004) | 1.014 (0.993,1.035) | 1.008 (0.997,1.019) | 68.17 |
| 10% | 1.000 (0.995,1.005) | 1.027 (0.999,1.055) | 1.015 (1.001,1.030) | 68.94 |
| 50% | 1.000 (0.990,1.010) | 1.136 (1.088,1.184) | 1.070 (1.042,1.097) | 72.80 |
| 99% | 1.000 (0.987,1.014) | 1.268 (1.268,1.304) | 1.122 (1.090,1.153) | 76.49 |

[a] Valid MR-GxE is used to indicate MR-GxE estimates obtained using a single interaction covariate for which GxE2 is satisfied.

**Table 3. Simulated results using differing proportions of non-zero gene-by-covariate interaction with respect to the outcome (simulation 5).**

| Proportion $\beta_{4K} \neq 0$ | Valid MR-GxE[a]$\hat{\beta}_1$ (95% CI) | Mean MR-GxE $\hat{\beta}_1$ (95% CI) | MR-GENIUS $\hat{\beta}_1$ (95% CI) | Mean F-statistic |
|---|---|---|---|---|
| 0% | 1.000 (0.998,1.002) | 1.000 (0.990,1.010) | 1.000 (0.995,1.005) | 67.62 |
| 1% | 1.000 (0.998,1.002) | 1.004 (0.990,1.017) | 1.001 (0.995,1.008) | 67.64 |
| 5% | 1.001 (0.997,1.004) | 1.019 (0.995,1.042) | 1.008 (0.997,1.019) | 67.66 |
| 10% | 1.000 (0.995,1.005) | 1.036 (1.004,1.069) | 1.016 (1.001,1.031) | 67.69 |
| 50% | 1.000 (0.990,1.010) | 1.185 (1.117,1.254) | 1.082 (1.054,1.110) | 67.52 |
| 99% | 0.997 (0.987,1.014) | 1.366 (1.300,1.432) | 1.163 (1.131,1.194) | 67.63 |

[a]Valid MR-GxE is used to indicate MR-GxE estimates obtained using a single interaction covariate for which GxE3 is satisfied.

simulated examples. Initially, we generated data as in simulations 3–4, instead varying the proportion of candidate interactions with non-zero second stage interactions ($\beta_{4k} \neq 0$) (simulation 5). The first-stage interaction coefficient across all interaction was set to $\gamma_{3k} = 1$, with designated invalid interactions having a second stage interaction($\beta_{4k} = 1$). For each proportion the mean MR-GENIUS estimate was obtained, as well as the mean MR-GxE estimate across all candidate interactions. In addition, a mean MR-GxE estimate using a single randomly sampled interaction for which $\beta_4 = 0$ is also provided, with results presented in Table 3.

In this scenario it can be seen that MR-GxE and MR-GENIUS exhibit bias in the direction of GxE2 violation, while utilising a single interaction for which GxE3 is satisfied provides unbiased estimates. This would suggest that MR-GENIUS is reliant upon $\bar{\beta}_{4k} = 0$ across all implicitly leveraged interactions.

To further demonstrate the impact of GxE3 violation, as well as the utility of employing an adapted Sargan test as a sensitivity analysis, we present a further simulation shown in Table 4 (simulation 6). In this case, a score analogous to a PRS was used as a single IV, comprised of 1,000 individual sub-instruments of approximately equal strength. Mirroring the previous simulated example, the true causal effect was defined as $\beta_1 = 1$ with a horizontal pleiotropic effect $\beta_2 = 0.05$. Sub-instruments violating assumption GxE3 were estimated to have a value $\beta_4 = 0.2$, varying the proportion of invalid sub-instruments. The mean F-statistic across all iterations simulations was 98.80 (Breusch-Pagan 31.80, $p-value = 0.013$), and MR-GENIUS estimates are presented for comparison.

As shown in Table 4, both MR-GxE and MR-GENIUS produce biased causal effect estimates when the constant pleiotropy assumption is violated. Violation of the constant pleiotropy assumption is also detected by applying a Sargan test, provided all sub-instruments do

**Table 4. Simulated results illustrating use of Sargan test to identify GxE3 violation (simulation 6).**

| Proportion[a] $\hat{\beta}_{4k} \neq 0$ | MR-GxE $\hat{\beta}_1$ (95% CI) | MR-GENIUS $\hat{\beta}_1$ (95% CI) | Mean F-statistic | Sargan p-value |
|---|---|---|---|---|
| 0% | 1.000 (0.997, 1.003) | 1.000 (0.963,1.036) | 98.95 | 0.480 |
| 1% | 1.010 (1.006, 1.014) | 1.005 (0.956,1.054) | 98.95 | <0.001 |
| 5% | 1.050 (1.045, 1.056) | 1.022 (0.940,1.104) | 98.88 | <0.001 |
| 10% | 1.100 (1.093, 1.110) | 1.049 (0.937,1.161) | 98.92 | <0.001 |
| 50% | 1.499 (1.485. 1.514) | 1.256 (1.012,1.551) | 98.95 | <0.001 |
| 100% | 2.000 (1.981, 2.020) | 1.505 (1.149,1.861) | 98.77 | 0.456 |

[a]Proportion $\beta_{4K} \neq 0$ refers to the proportion of sub-instruments which violate assumption GxE3.

not identically violate GxE3. As the Sargan test relies upon at least one instrument being valid, identical violation of assumption GxE3 would also violate the assumptions of the conventional Sargan approach.

## Estimating the effect of adiposity on systolic blood pressure within the UK Biobank

To demonstrate each of the sensitivity analyses previously described, we performed MR analyses estimating the causal effect of adiposity (measured using BMI) on SBP using data from the UK Biobank. The UK Biobank obtained written consent from all participants, and received ethical approval from the Research Ethics Committee (REC reference for UK Biobank is 11/NW/0382). This analysis was approved by the UK Biobank access committee as part of project 8786. Consent was sought by UK Biobank as part of the recruitment process. This serves as a re-examination of the original applied example in Spiller et al. (2019) who first proposed the MR-GxE model [8]. The UK Biobank has approval from the North West Multi-centre Research Ethics Committee (MREC) as a Research Tissue Bank (RTB) approval, and consequently separate ethical clearance was not required for this project which was conducted under the RTB approval. In this study we evaluate each underlying assumption using the diagnostic tools described above, and contrasting the results with MR-GENIUS [8, 13]. After performing quality control, removing participants with missing data, and restricting the sample to unrelated individuals of European ancestry, a total of 358, 928 participants were included in the analyses.

MR-GxE was implemented by constructing a weighted PRS informed using genetic variants previously identified from the GIANT consortium [22]. As the GIANT consortium represents a subset of the most recent UK Biobank release, subsequent analyses have been conducted in a one-sample framework. A total of 95 independent genetic variants were used after performing linkage disequilibrium (LD) pruning, and removing tri-allelic or palindromic variants. Finally, we standardized BMI, SBP, and the weighted PRS using a z-score transformation prior to performing analyses. In previous work we found evidence of a positive association between BMI and SBP using OLS and TSLS regression approaches [8, 23–25].

Initially, a discovery subset ($N = 100, 000$) was randomly sampled from the UK Biobank data for use in identifying interactions for MR-GxE analyses. Causal effect estimates and sensitivity analyses were performed using the remaining data. Candidate gene-by-covariate interactions were detected by estimating the first-stage F-statistic for 576 candidate interaction covariates within the UK Biobank. After applying a multiple testing correction, the 20 interaction covariates with the strongest association were selected and utilised in subsequent analyses. Table 5 shows MR-GxE estimates of causal effect and corresponding sensitivity analyses with respect to each interaction covariate. The strength of each interaction across the set of candidate interaction covariates is illustrated in Fig 6, where annotations give the UK Biobank field ID for each interaction covariate.

To assess assumption GxE3, we created 9 sub-instruments sampling from the 95 SNPs used to create the initial PRS instrument. Fitting the MR-GxE model using multiple sub-instruments allows for over identification tests to be performed, testing the extent to which causal effect estimates differ when using individual sub-instruments. In each case, a failure to reject the null can be considered to be evidence of interaction exogeneity as previously outlined. To implement this approach, the set of SNPs were randomly assorted into 9 sub-instruments of approximately equal strength, quantified using the F-statistic with respect to BMI. Repeating this procedure using sub-instruments containing differing SNPs yielded similar results. We also present the mean F-statistic across the set of sub-instruments to emphasise their strength.

**Table 5. MR-GxE estimates and sensitivity analyses using each candidate interaction covariate and MR-GENIUS.**

| Covariate (UK Biobank Field ID) | F-Statistic | $\hat{\beta}_1$ (p-value) | $\rho(G, Z)$[a] (p-value) | Sargan[b] (p-value) | Mean F[c] |
|---|---|---|---|---|---|
| Waist circumference (f.48.0.0) | 182.86 | -0.524 (<0.001) | 0.103 (<0.001) | 7.531 (0.481) | 79.40 |
| Weight (kg) (f.21002.0.0) | 123.16 | -0.687 (<0.001) | 0.119 (<0.001) | 9.342 (0.314) | 48.79 |
| Diabetes diagnosis (f.2443.0.0) | 54.22 | -0.065 (0.470) | 0.020 (<0.001) | 12.19 (0.143) | 41.22 |
| Alcohol intake frequency (f.1558.0.0) | 50.65 | 0.163 (0.006) | 0.001 (0.526) | 5.69 (0.682) | 41.62 |
| Physical activity (vigorous) (f.904.0.0) | 42.10 | 0.017 (0.862) | 0.004 (0.003) | 20.40 (0.009) | 17.03 |
| Vascular/ heart problem diagnosis (f.6150.0.0) | 33.65 | -0.446 (<0.001) | 0.028 (<0.001) | 7.22 (0.513) | 16.76 |
| Time number displayed during memory test (f.4253.0.5) | 28.42 | -2.155 (0.333) | 0.015 (0.002) | 14.54 (0.069) | 13.51 |
| Number of days per week walked 10+ mins (f.864.0.0) | 27.87 | 0.208 (0.011) | 0.001 (0.705) | 6.87 (0.551) | 18.98 |
| DBP (automated, baseline) (f.4079.0.0) | 26.45 | -0.324 (<0.001) | 0.020 (<0.001) | 6.39 (0.603) | 16.32 |
| Physical activity (moderate) (f.884.0.0) | 23.60 | 0.165 (0.107) | 0.001 (0.324) | 3.66 (0.886) | 14.27 |
| Townsend deprivation index (f.189.0.0) | 23.01 | 0.108 (0.489) | -0.016 (<0.001) | 9.24 (0.323) | 16.80 |
| Comparative body size at age 10 (f.1687.0.0) | 20.65 | 0.283 (0.004) | 0.048 (<0.001) | 9.94 (0.269) | 14.62 |
| Time to complete pair matching activity (f.400.0.2) | 20.49 | 0.052 (0.689) | -0.007 (<0.001) | 29.50(< 0.001) | 11.84 |
| Pulse rate (f.4194.0.0) | 20.45 | 0.031 (0.873) | -0.010 (<0.001) | 13.78 (0.088) | 4.77 |
| Time watching television (f.1070.0.0) | 20.01 | -0.140 (0.211) | 0.017 (<0.001) | 14.83 (0.063) | 15.08 |
| DBP (automated, follow-up) (f.4079.0.1) | 19.55 | -0.501 (<0.001) | 0.016 (<0.001) | 6.37 (0.606) | 11.72 |
| Own or rent accommodation (f.680.0.0) | 18.41 | 0.078 (0.607) | -0.006 (<0.001) | 19.84 (0.011) | 10.13 |
| Age at assessment (f.21003.0.0) | 18.15 | 0.697 (<0.001) | 0.013 (<0.001) | 17.28 (0.027) | 14.03 |
| Birthweight known (f.120.0.0) | 17.93 | 0.067 (0.851) | -0.014 (<0.001) | 14.16 (0.078) | 4.94 |
| Year of birth | 15.85 | 0.710 (<0.001) | -0.014 (<0.001) | 17.12 (0.029) | 13.94 |
| **OLS** | - | **0.186 (<0.001)** | - | - | - |
| **TSLS** | **7776.52** | **0.130 (<0.001)** | - | - | - |
| **MR-GENIUS** | **1332.7 (<0.001)[d]** | **0.034 (0.009)** | - | - | - |

[a] $\rho(G, Z)$ represents the correlation between the PRS and interaction covariate,

[b] Sargan shows the results to over identification tests using sub-instruments,

[c] The mean F-statistic for sub-instruments,

[d] BP Heterogeneity Test.

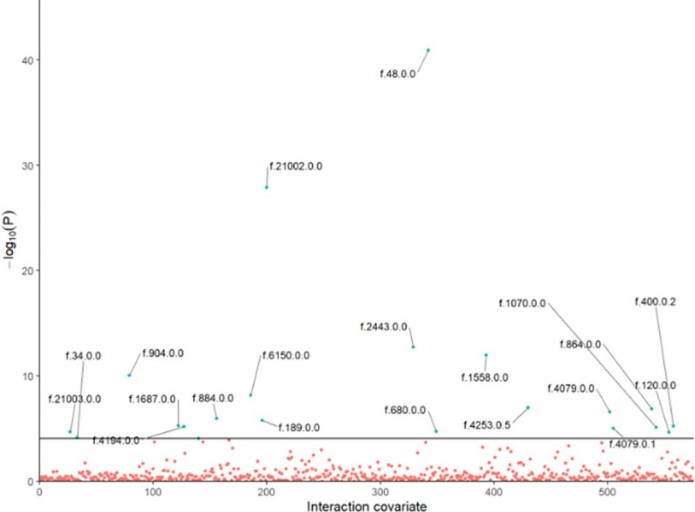

**Fig 6. Identified gene-by-covariate interactions with respect to genetically predicted body mass index.** A scatter plot showing the first-stage F-statistics for instrument-by-covariate interactions using data from UK Biobank. A horizontal line is included representing the Bonferroni correction for statistical significance. For clarity, blue points represent interactions identified after multiple testing. The 20 strongest interactions have been annotated using their UK Biobank field identification number.

As shown in Table 5, there exists substantial disagreement across the range of selected inter-action covariates, suggesting that one or more violate underlying assumptions of the MR-GxE approach.

Considering assumption GxE2, several of the identified gene-by-covariate interactions are proxy measures of adiposity, specifically waist circumference, weight in kilograms, and com-parative body size at age 10. Such interaction covariates are often problematic, as associations between the genetic variants and the interaction can result in collider bias where the interac-tion covariate is downstream of the exposure (see Materials and methods). In this case, higher estimates of $\rho(G, Z_k)$ for these variables supports this interpretation, and their subsequent exclusion from further analyses. A similar argument can also be made with respect to interac-tion covariates downstream of BMI, including diabetes diagnosis, vascular/heart problem diagnosis, and diastolic blood pressure (DBP).

By applying Sargan tests, a number of interaction covariates related to cognition, physical activity, and age appear to violate assumption GxE3. This could be explained by the gene-by-covariate interactions relating to one or more underlying risk factors, which are not adjusted for in the corresponding MR-GxE models.

After applying sensitivity analyses, three interaction covariates can be identified as appro-priate choices for estimation using MR-GxE. This selection was made using Sargan test and correlation p-value thresholds of p-value<0.0025, applying a multiple testing correction. Selected covariates include alcohol intake frequency and physical activity, both days walked and moderate levels of exercise. Considering alcohol intake and physical activity, the lack of a substantial correlation between each interaction covariate and the PRS suggests that violation of GxE2 is unlikely.

In previous work Townsend deprivation index (TDI) was selected as an interaction covari-ate in a summary MR-GxE analysis and returned estimates in agreement with both alcohol consumption and physical activity measures identified above. However, it is important to note that TDI shows evidence of a non-zero instrument-interaction covariate correlation,

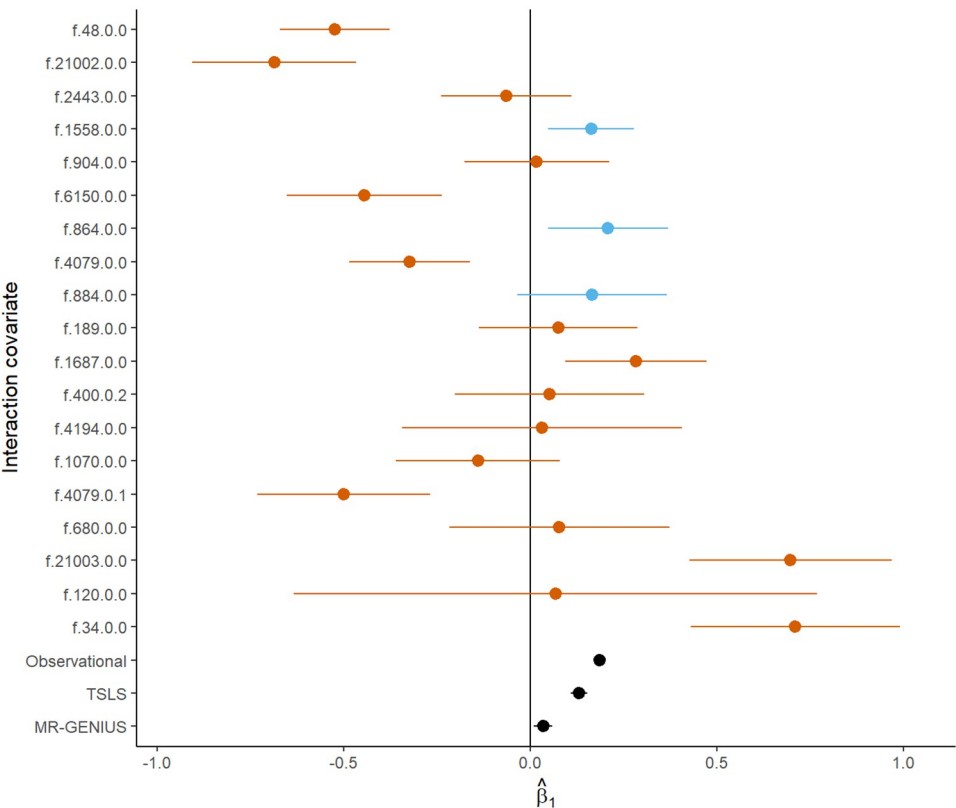

**Fig 7. A forest plot showing MR-GxE causal effect estimates using the interaction covariates presented in Table 5.**
Observation f.4253.0.5 has been omitted for clarity. Red points indicate analyses for which assumptions may likely be violated, while blue points show potentially valid interaction covariates using accompanying sensitivity analyses. Observational, two-stage least squares (TSLS), and MR-GENIUS estimates are also shown as black points.

potentially highlighting a violation of assumption GxE2. This can be explained by TDI being plausibly downstream of both BMI and the instrument, representing situation in which the correlation does not invalidate estimates of causal effect.

Crucially, adopting alcohol and physical activity as interaction covariates yields causal effect estimates which appear biologically plausible, and support evidence from both observational and MR studies suggesting a positive association between BMI and SBP. Estimates using each interaction covariate are presented in Fig 7.

As a final analysis, we implemented MR-GENIUS using the PRS, BMI, and SBP measures from UK Biobank. This resulted in a more precise estimate in comparison to MR-GxE, however, the effect estimate appears to strongly disagree with evidence from MR-GxE and alternate approaches. Given MR-GENIUS implicitly relies upon analogous assumptions to MR-GxE, it seems reasonable to assume that such a discrepancy could arise from bias due to violations stemming from one or more unmeasured interactions. This is further supported by MR-GxE estimates of similar direction and magnitude which appear to show evidence of bias, such as vigorous physical activity which shows evidence of GxE3 violation.

## Discussion

In this paper we examine two related interaction-based MR approaches: MR-GxE and MR-GENIUS. Both MR-GxE and MR-GENIUS rely upon similar underlying assumptions,

whilst differing based on whether a gene-by-covariate interaction needs to be explicitly incorporated within the estimation model. Specifically, MR-GxE relies upon at least a single measured gene-by-covariate interaction which satisfies assumptions GxE 1–3, whilst MR-GENIUS does not require such an interaction to be observed. However, as a consequence of implicitly leveraging multiple underlying interactions, the MR-GENIUS approach requires assumptions GxE 1–3 to hold globally. Essentially, stronger assumptions are required to mitigate the absence of an observed gene-by-covariate interaction. It should be emphasised however, that evaluation of MR-GENIUS in this paper does not consider the inclusion of observed gene-by-covariate interactions.

Through an examination of the MR-GxE assumptions, several approaches aiming to evaluate assumptions GxE 1–3 have been outlined. Interaction strength (GxE1) can be evaluated using the first-stage F-statistic for the interaction term, analogous to evaluating instrument strength in conventional MR. The corresponding global test for interaction strength using MR-GENIUS and a continuous exposure is the Breusch-Pagan test for heteroskedasticity [13–16].

Assumption GxE2 can initially be evaluated by estimating the correlation between $Z_i$ and both $G_i$ and $X_i$ respectively. Where $Z_i$ is observed to be correlated with $G_i$, it is possible that a confounding relationship exists violating assumption GxE2. Further, the simultaneous association of $Z_i$ with $G_i$ and $X_i$ can result in bias where $Z_i$ is downstream of $X_i$. However, as the existence of such correlations does not necessarily imply that this assumption is violated, a more promising approach may be to adopt an interaction covariate $Z_i$ which is highly likely to be exogenous (see Materials and methods). For example, one could employ genetic variants which instrument a likely interaction covariate. Future work will explore this possibility.

The constant pleiotropy assumption (GxE3) can be tested in cases where the initial instrument $G_i$ is a composite instrument, that is, comprised of multiple sub-instruments such as genetic variants within a PRS. Heterogeneity in effect estimates obtained using sub-instruments can be considered as evidence of violation of the constant pleiotropy assumption, analogous to heterogeneity in two-sample summary MR [7, 26]. In principle, a similar approach can be applied using sub instruments with MR-GENIUS, though such an examination is beyond the scope of this paper. A summary of the MR-GxE assumptions and proposed tests is given in Table 6.

In the applied analysis the effect of BMI on SBP was estimated using MR-GENIUS and a range of interaction covariates in conjunction with MR-GxE. We identified three suitable interaction covariates, which suggest a positive effect of BMI upon SBP in agreement with previous observational and MR analyses. Importantly, we highlight interaction covariates which

**Table 6. A summary of the MR-GxE assumptions and proposed sensitivity analyses.**

| Assumption | Description | Consequence of violation | Tool to assess plausibility |
|---|---|---|---|
| Interaction strength (GxE1) | An observed gene-by-covariate interaction $GZ$ should be selected, such that the association between the instrument $G$ and exposure $X$ varies across levels of $Z$ | Insufficient precision to detect causal effects and directional bias when multiple interactions are used. | Estimating the first stage F-statistic for $GZ$ and adopting an interaction covariate such that $F \geq 10$. |
| Interaction exogeneity (GxE2) | The gene-by-covariate interaction $GZ$ should be independent of confounders of the exposure $X$ and outcome $Y$. | Inflated type-I error rates when evaluating instrument validity and biased effect estimates for the effect of $X$ on $Y$. | Estimating the association between the instrument $G$ and interaction covariate $Z$, selecting an interaction such that $G$ and $Z$ are independent. |
| Constant pleiotropy (GxE3) | The direct effect of an instrument $G$ on the outcome $Y$ should remain constant across levels of the interaction covariate $Z$. | Estimates of the effect of the exposure $X$ on $Y$ will be biased in the direction of the effect of $GZ$ on $Y$. | Using a Sargan test when sub-instruments can be constructed from a composite instrument $G$. |

violate the MR-GxE assumptions and link these issues to the possibly biased effect estimates obtained using MR-GENIUS.

Several limitations remain with respect to MR-GxE which warrant further explanation. Firstly, reliance upon an observed gene-by-covariate interaction limits the extent to which the method can be applied in contrast to MR-GENIUS. We advocate the use of MR-GENIUS in cases where no interaction covariate is available, though care needs to be taken in justifying the more stringent assumptions MR-GENIUS entails if an interaction covariate is not specified. Second, evaluating GxE2 using the correlation of between $Z_i$ and $G_i$ does not provide a clear indication of whether the assumptions hold. It is possible that GxE2 can be violated when $Z_i$ and $G_i$ appear to be independent, and assuming the direction of effect between $Z_i$ and $X_i$ relies upon a priori knowledge regarding the direction of association. It is therefore critical to identify plausible biological mechanisms underpinning the observed relationships in the MR-GxE model.

Finally, whilst an overidentification test has been presented for evaluating GxE3, there is not at present a method aiming to correct for violation of the constant pleiotropy assumption. It is likely that pleiotropy robust methods, such as median or modal regression, could be utilised to correct for resulting bias, and the application of such methods will be fully explored in future work.

## Conclusion

MR-GxE and MR-GENIUS are two interaction-based MR approaches which leverage gene-by-covariate interactions to estimate causal associations, while correcting for instrument invalidity. MR-GxE can be adapted to the individual level data setting and allows for the underlying assumptions of the approach to be tested provided a gene-by-covariate interaction is explicitly identified. In contrast, MR-GENIUS does not require such an interaction to be identified, but instead relies upon a more stringent set of assumptions analogous to MR-GxE. The use of each method should therefore reflect the specific research questions considered, as each approach is especially suited to particular research contexts. However, it is essential that the strengths and limitations of each approach are given sufficient consideration prior to their application.

## Supporting information

**S1 Table. Simulated results and effect estimates for subset of interaction (denoted *Z*) identified from Fig 5A (simulation 1).**
(PDF)

## Acknowledgments

The authors would like to thank participants of the UK Biobank from which individual-level data was sourced. Analyses were performed under application number 8786.

## Author Contributions

**Conceptualization:** Wes Spiller, Jack Bowden.

**Formal analysis:** Wes Spiller.

**Investigation:** Wes Spiller.

**Methodology:** Wes Spiller, Jack Bowden.

**Supervision:** George Davey Smith, Jack Bowden.

**Writing – original draft:** Wes Spiller.

**Writing – review & editing:** Wes Spiller, Fernando Pires Hartwig, Eleanor Sanderson, George Davey Smith, Jack Bowden.

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
