## [Decision Letter · Decision Letter 0]

26 Apr 2022

PONE-D-22-06423Interaction-based Mendelian randomization with measured and unmeasured gene-by-covariate interactionsPLOS ONE

Dear Dr. Spiller:

Thank you for submitting your manuscript to PLOS ONE. After careful consideration, we feel that it has merit but does not fully meet PLOS ONE’s publication criteria as it currently stands. Therefore, we invite you to submit a revised version of the manuscript that addresses the points raised during the review process. Best wishes, Momiao Xiong

We look forward to receiving your revised manuscript.

Kind regards,

Momiao Xiong

Academic Editor

PLOS ONE

Journal Requirements:

"Wes Spiller is supported by a Wellcome Trust studentship (108902/B/15/Z)."

Reviewers' comments:

Reviewer's Responses to Questions

**Comments to the Author**

1. Is the manuscript technically sound, and do the data support the conclusions?

Reviewer #1: Yes

Reviewer #2: Yes

2. Has the statistical analysis been performed appropriately and rigorously? 

Reviewer #1: Yes

Reviewer #2: Yes

3. Have the authors made all data underlying the findings in their manuscript fully available?

Reviewer #1: Yes

Reviewer #2: Yes

4. Is the manuscript presented in an intelligible fashion and written in standard English?

Reviewer #1: Yes

Reviewer #2: Yes

5. Review Comments to the Author

Reviewer #1: The paper discusses the relative strengths and limitations of interaction-MR approaches for both MR-GxE and MR-GENIUS methods, which is a great contribution to the field. I have a few comments below:

1. For simulation settings, could you please provide more details how the variables in equations are generated (i.e. what are the distributions they come from and the associated parameters)?

2. For simulation results, could you make it clear what type of outcomes they are (seems the results are for continuous exposure and outcome).

3. Have you tried to extend the simulations to other types of outcomes (i.e. binary outcome)?

Thanks.

Reviewer #2: Spiller et al explore the performance of two Mendelian randomization approaches that rely on genotype x environment interactions (MR-GxE and MR-GENIUS) to correct for the pervasive problem of genetic pleiotropy in ordinary MR analyses. The authors review both methods, propose a new formulation of MR-GxE in the two stage least squares framework, propose a series of sensitivity analyses to evaluate the robustness of the methods to violations of underlying assumptions, and then finally explore the (known) causal relationship between BMI and systolic blood pressure in the UK Biobank.

General comments:

-This is an exceptionally well written manuscript that I thoroughly enjoyed reading. The background material in the Materials and Methods was very useful - providing intuition for understanding the mechanics of the approaches and without sacrificing intellectual rigour. I do not have any major concerns with the paper, but do have a number of suggestions for improving its clarity in certain parts, and some additional points that the author might want to consider.

Line 110 Page 5: The notation is confusing here. I assume this is meant to be epsilon_hat_subscript_X_i but it could also be read as epsilon_hat multiplied by X_subscript_i.

Line 138 Page 5: “First, when using a single interaction the F-statistic cannot be related to the magnitude of relative bias, as at least three instruments would be required for the asymptotic formula to be valid.”. This sentence is not clear to me.

Top of page 6: Scale dependency- obviously scale dependent interactions may also be present in the absence of transformation (i.e. when the variables are on their original scale).

Line 159 Page 6: “Specifically, MR-GENIUS relies upon the residual error in a regression of the exposure upon the genetic IV to be heteroskedastic, such that (X|G) = E(ϵ2X|G) [13]. This is evaluated using a Breusch-Pagan test for heteroskedasticity [13].” Again, I find this a little unclear especially the equation. My understanding is that in a Breusch-Pagan test the square of the first stage residuals is regressed on the IV (here genotype). I assume you are saying that the expected value of the squared residuals given G doesn’t change across levels of G- but is this the correct way to say this using an equation?

Line 163-171. I assume that these analyses to identify gene x covariate interactions are based on polygenic risk scores, rather than individual variants. I suggest you make this clear, especially since single variants are unlikely to show large GxE for most traits.

Line 173 (page 6) “In previous work we show how assumption GxE2 is potentially violated when certain confounding structures exist, specifically, where Gi and Zki are simultaneously downstream of a confounder Ui or where there is an open path between the two variables through Ui”. Could the authors please discuss the intuition for this result perhaps with the aid of a path diagram (or description)? Do you need both paths to be present (i.e. a path from U to G AND a path from U to Z, or is only one sufficient to produce bias?). Why? Maybe this could be pointed out to the reader?

“Relating assumption GxE2 to the MR-GENIUS approach, associations violating GxE2 would imply associations vary across values of the unmeasured confounders violating the second MR-GENIUS assumption [13]. However, this problem can be mitigated by incorporating additional interaction covariates within the MR-GENIUS model, as described in Tchetgen Tchetgen et al, 2021 [13].” Please expand on this.

“The third MR-GxE assumption requires pleiotropic effects of Gi upon Yi to remain

constant across values of Zki, with the gene-by-covariate interaction being independent

of Yi when conditioning on Xi.”. I would put in parentheses after this statement (i.e. beta_4 equals zero)

Some further thoughts:

A table summarizing the core assumptions, the consequences for violating them, and methods of testing them might be useful for readers.

One of the difficulties in ordinary MR studies is whether the SNPs chosen to be instruments are primarily associated with the exposure, or the outcome. Various methods (e.g. Steiger filtering) have been proposed to get a handle on this problem. My question is whether the addition of SNP*covariate terms in GZ where the SNP is primarily an outcome associated SNP is problematic? My guess would be yes (it implies beta_4 is not null). Do the authors think that this would be a common problem? If so, could they recommend procedures to guide against this possible source of bias?

Sex seems an obvious candidate for a genotype x “environment” interaction. Was this included in the list of covariates and results not presented because the evidence for interaction was so low?

You have a “positive empirical control” in this paper (i.e. we know BMI causes SBP) would it be worth also including a negative empirical control? (i.e. two phenotypes where we are pretty sure the exposure doesn’t cause the outcome). I realize that this may potentially be a substantial amount of work so would be happy for the paper to be accepted without it, but i think it is something worth considering.

6. PLOS authors have the option to publish the peer review history of their article (what does this mean?). If published, this will include your full peer review and any attached files.

Reviewer #1: No

Reviewer #2: No

---

## [Author Response · Author response to Decision Letter 0]

3 Jun 2022

Please see the formatted letter submitted with the manuscript. The contents of the letter are also included below.

Many thanks,

Response to reviewers

Dear Reviewers,

Many thanks for your very kind and careful consideration of our manuscript. Please find responses to each of your comments below, which we hope will explaining how we have attempted to address the issues highlighted.

Journal requirements

Thank you for highlighting these potential issues. The author summary has been removed, and the file names have been changed to correspond to the journal guidelines.

I do not believe we have cited any retracted papers, and to my knowledge all papers have been cited correctly.

Additional information as been provided regarding written consent obtained from each participant. The study is conducted using data collected by the UK Biobank, and no additional data was obtained for this manuscript.

4. Thank you for stating the following financial disclosure: "Wes Spiller is supported by a Wellcome Trust studentship (108902/B/15/Z)."Please state what role the funders took in the study. If the funders had no role, please state: "The funders had no role in study design, data collection and analysis, decision to publish, or preparation of the manuscript."

The suggested statement has been included in the funding section for resubmission.

5. In your Data Availability statement, you have not specified where the minimal data set underlying the results described in your manuscript can be found. PLOS defines a study's minimal data set as the underlying data used to reach the conclusions drawn in the manuscript and any additional data required to replicate the reported study findings in their entirety. All PLOS journals require that the minimal data set be made fully available. For more information about our data policy, please see http://journals.plos.org/plosone/s/data-availability. Upon re-submitting your revised manuscript, please upload your study’s minimal underlying data set as either Supporting Information files or to a stable, public repository and include the relevant URLs, DOIs, or accession numbers within your revised cover letter. For a list of acceptable repositories, please see http://journals.plos.org/plosone/s/data-availability#loc-recommended-repositories. Any potentially identifying patient information must be fully anonymized. Important: If there are ethical or legal restrictions to sharing your data publicly, please explain these restrictions in detail. Please see our guidelines for more information on what we consider unacceptable restrictions to publicly sharing data: http://journals.plos.org/plosone/s/data-availability#loc-unacceptable-data-access-restrictions. Note that it is not acceptable for the authors to be the sole named individuals responsible for ensuring data access. We will update your Data Availability statement to reflect the information you provide in your cover letter.

Many thanks for highlighting potential issues surrounding data availability. As stated in the data availability section, the data used in simulations is available from https://github.com/WSpiller/GxE_Simulation and the code produces the data sets evaluated for each simulation.

The UK Biobank data unfortunately cannot be made publicly available following the legal requirements for access to the data. We have included the application number under which the analyses were performed, in addition to a link to information for researchers who wish to apply for access. Once access is granted through the UK Biobank system, researchers will have the necessary data to replicate our analyses.

Reviewer comments

Reviewer #1

The paper discusses the relative strengths and limitations of interaction-MR approaches for both MR-GxE and MR-GENIUS methods, which is a great contribution to the field.

Many thanks for your kind words regarding our manuscript.

1. For simulation settings, could you please provide more details how the variables in equations are generated (i.e. what are the distributions they come from and the associated parameters)?

Thank you for highlighting this issue. Additional information has now been added to page 3, describing how each variable is generated. Each variable is treated as continuous, with each observation either randomly sampled from a normal distribution with mean 0 and standard deviation of 1, or as a combination of prespecified covariates and an error term randomly sampled from a normal distribution with mean 0 and standard deviation of 1.

2. For simulation results, could you make it clear what type of outcomes they are (seems the results are for continuous exposure and outcome).

This is tied with the previous comment, where there was some ambiguity regarding the generated variables. We have now stated that all variables are continuous for simplicity. Thank you for highlighting this oversight.

3. Have you tried to extend the simulations to other types of outcomes (i.e. binary outcome)?

This is a very interesting question, as often binary outcomes are of interest in clinical settings. In the most simple setting, a researcher could fit the linear model to a binary exposure using a linear probability model, though that can have it’s own problems (interpretability, predicted values less than 0 or greater than 1 etc.)

We believe that the MR-GxE model could be adapted into a two-stage predictor substitution model, which would essentially use a first stage linear model, and then incorporate the fitted values within a second stage logistic regression model. There is no reason to believe this wouldn’t work, but there are two reasons why we haven’t explored this possibility in the paper.

First, MR-GENIUS is typically applied to continuous outcomes, while allowing for binary instruments and exposures. As one of the main focuses of the paper is a comparison between the MR-GxE and MR-GENIUS approaches, it was thought best to conduct all analyses within the continuous outcome setting. Second, we wanted to keep the manuscript to an appropriate length for publication, and evaluating binary outcome MR-GxE models with sufficient detail would increase the length of the paper substantially. As the potential issues and body of work related to binary outcomes in IV analyses (and binary exposures), we thought it best that a detailed discussed be deferred to future work.

 

Reviewer #2

This is an exceptionally well written manuscript that I thoroughly enjoyed reading. The background material in the Materials and Methods was very useful - providing intuition for understanding the mechanics of the approaches and without sacrificing intellectual rigour. I do not have any major concerns with the paper, but do have a number of suggestions for improving its clarity in certain parts, and some additional points that the author might want to consider.

Thank you for your very kind words and comments on the paper.

Line 110 Page 5: The notation is confusing here. I assume this is meant to be epsilon_hat_subscript_X_i but it could also be read as epsilon_hat multiplied by X_subscript_i.

You are correct, this is a typo which we have now amended.

Line 138 Page 5: “First, when using a single interaction the F-statistic cannot be related to the magnitude of relative bias, as at least three instruments would be required for the asymptotic formula to be valid.”. This sentence is not clear to me.

Thank you for highlighting the need for greater clarification on this point. In conventional IV analyses there is great interest in the consequences of using weak instruments to estimate causal effects. In MR where we can often have multiple instruments, it is possible to determine the direction of weak instrument bias based on whether instrument-exposure and instrument-outcome associations are obtained from non-overlapping samples. This is true, however, only in instances where multiple instruments (or for MR-GxE interactions) are used.

Admittedly this is more of a technical point however, as the use of weak instruments will typically reduce precision to the point that the direction of weak instrument bias is of marginal interest. It is something we mention because it is highlighted as a useful feature of MR with multiple instruments in the two sample setting (producing more conservative effect estimates). The section has now been rewritten to improve readability.

Top of page 6: Scale dependency- obviously scale dependent interactions may also be present in the absence of transformation (i.e. when the variables are on their original scale).

This is absolutely true. There is no guarantee that the interaction observed is not an artifact of the arbitrary means with which it is initially measured. This is often why the most convincing GxE interactions are identified using a combination of sources, and why surprising GxE interactions from large-scale interrogation of studies with genetic data warrant careful consideration before being used.

Line 159 Page 6: “Specifically, MR-GENIUS relies upon the residual error in a regression of the exposure upon the genetic IV to be heteroskedastic, such that (X|G) = E(ϵ2X|G) [13]. This is evaluated using a Breusch-Pagan test for heteroskedasticity [13].” Again, I find this a little unclear especially the equation. My understanding is that in a Breusch-Pagan test the square of the first stage residuals is regressed on the IV (here genotype). I assume you are saying that the expected value of the squared residuals given G doesn’t change across levels of G- but is this the correct way to say this using an equation?

This is an excellent point to make, as we have tried to present instrument strength with regard to MR-GENIUS as close to the representation in their paper as possible. We agree in hindsight that the equation is more confusing than helpful, and so have adjusted this section to highlight that the residual error from regressing X on G should be heteroskedastic, and that a BP-Test can be used to evaluate this criterion.

Line 163-171. I assume that these analyses to identify gene x covariate interactions are based on polygenic risk scores, rather than individual variants. I suggest you make this clear, especially since single variants are unlikely to show large GxE for most traits.

Thank you for the suggestion, and we completely agree. Often it is most appropriate to use a PRS as an instrument G, and we have now stated this explicitly in the paper.

Line 173 (page 6) “In previous work we show how assumption GxE2 is potentially violated when certain confounding structures exist, specifically, where Gi and Zki are simultaneously downstream of a confounder Ui or where there is an open path between the two variables through Ui”. Could the authors please discuss the intuition for this result perhaps with the aid of a path diagram (or description)? Do you need both paths to be present (i.e. a path from U to G AND a path from U to Z, or is only one sufficient to produce bias?). Why? Maybe this could be pointed out to the reader?

We agree that the description of interaction exogeneity needed greater attention, and thank the reviewer for highlighting this issue. We have edited this section to better reflect issues related to confounding in the context of MR-GxE. This was a particularly interesting comment, as it links to the use of MR-GxE as a test for instrument validity in a way not communicated in the paper. Many thanks.

“Relating assumption GxE2 to the MR-GENIUS approach, associations violating GxE2 would imply associations vary across values of the unmeasured confounders violating the second MR-GENIUS assumption [13]. However, this problem can be mitigated by incorporating additional interaction covariates within the MR-GENIUS model, as described in Tchetgen Tchetgen et al, 2021 [13].” Please expand on this.

We have now included a description of how such a model would be constructed, though expanded MR-GENIUS model does not receive much attention in the original cited paper.

“The third MR-GxE assumption requires pleiotropic effects of Gi upon Yi to remain constant across values of Zki, with the gene-by-covariate interaction being independent of Yi when conditioning on Xi.”. I would put in parentheses after this statement (i.e. beta_4 equals zero)

We agree this additional clarification approves readability, and have made the change to the paper.

Some further thoughts:

A table summarizing the core assumptions, the consequences for violating them, and methods of testing them might be useful for readers.

This is an excellent suggestion. We have now included an additional table (Table 6) in the discussion which provides a brief overview of the assumptions, their consequences, and methods for evaluating their plausibility.

One of the difficulties in ordinary MR studies is whether the SNPs chosen to be instruments are primarily associated with the exposure, or the outcome. Various methods (e.g. Steiger filtering) have been proposed to get a handle on this problem. My question is whether the addition of SNP*covariate terms in GZ where the SNP is primarily an outcome associated SNP is problematic? My guess would be yes (it implies beta_4 is not null). Do the authors think that this would be a common problem? If so, could they recommend procedures to guide against this possible source of bias?

This is a common problem, and really highlights the care that needs to be taken when selecting SNPs which form the instrument G. We would recommend researchers use tools like Steiger filtering, as well as interrogate individual SNPs which are used to see if there are likely pleiotropic pathways (using a tool such as PhenoScanner for example). We agree that if the SNPs are primarily related to the outcome there is a greater chance for violation of GxE3, though this would require the effect on the outcome to also vary across levels of the interaction covariate Z. In terms of a procedure, assuming SNPs are selected as in conventional MR (F<10, Steiger filtering, using LD pruning) the only additional steps we would suggest at present would be to perform the tests described in the paper. This is somewhat unsatisfying as a recommendation, but anything more would require greater knowledge of the mechanisms underpinning the adopted GxE interaction, and if that is available then it would be more valuable to examine mechanisms through direct effects on of G on Y could vary across Z from a biological perspective.

Sex seems an obvious candidate for a genotype x “environment” interaction. Was this included in the list of covariates and results not presented because the evidence for interaction was so low?

This is correct, and often sex can serve as an ideal interaction covariate, owing to its plausible independence from genetic instruments which do not lie on sex chromosomes. There were two reasons why these interactions were omitted:

1. The GWAS effect estimates used to create the PRS for BMI were adjusted for participant sex, and so we wanted to avoid issues of inadvertently conditioning on a variable multiple times.

2. It is plausible that the underlying causal effect of BMI on SBP differs by biological sex, and ideally we rely on the underlying effect of BMI on SBP being the same across interaction subgroups (assumption IV4 of a homogenous causal effect applies to MR-GxE just as it does in conventional MR). The extent to which this would be the case is unclear, and a thorough examination of this issue would unfortunately lie outside the scope of this paper.

Even so, biological sex has been used in interaction MR analyses, most notably evaluating the impact of alcohol on cardiovascular disease using historic sex-stratified alcohol consumption in East Asian countries. The results from such studies are certainly plausible, and a deeper analyses of this (and IV4 in general) is something that would be exciting to explore.

You have a “positive empirical control” in this paper (i.e. we know BMI causes SBP) would it be worth also including a negative empirical control? (i.e. two phenotypes where we are pretty sure the exposure doesn’t cause the outcome). I realize that this may potentially be a substantial amount of work so would be happy for the paper to be accepted without it, but i think it is something worth considering.

This is a fantastic idea, and really brings the method back to its roots in negative controls (we ideally want an interaction covariate subgroup for which the instrument and exposure are independent). Unfortunately, given the number of simulations and the applied analysis it would be difficult to include the work while keeping the paper at an accessible length, though it is something we are keen to explore in the near future.

---

## [Decision Letter · Decision Letter 1]

11 Jul 2022

Interaction-based Mendelian randomization with measured and unmeasured gene-by-covariate interactions

PONE-D-22-06423R1

Dear Dr.Spiller ,

We’re pleased to inform you that your manuscript has been judged scientifically suitable for publication and will be formally accepted for publication once it meets all outstanding technical requirements.

Kind regards,

Momiao Xiong

Academic Editor

PLOS ONE

Additional Editor Comments (optional):

You have address all the issues  which the reviewers are concerned.

Reviewers' comments:

Reviewer's Responses to Questions

**Comments to the Author**

1. If the authors have adequately addressed your comments raised in a previous round of review and you feel that this manuscript is now acceptable for publication, you may indicate that here to bypass the “Comments to the Author” section, enter your conflict of interest statement in the “Confidential to Editor” section, and submit your "Accept" recommendation.

Reviewer #1: (No Response)

2. Is the manuscript technically sound, and do the data support the conclusions?

Reviewer #1: (No Response)

3. Has the statistical analysis been performed appropriately and rigorously? 

Reviewer #1: (No Response)

4. Have the authors made all data underlying the findings in their manuscript fully available?

Reviewer #1: (No Response)

5. Is the manuscript presented in an intelligible fashion and written in standard English?

Reviewer #1: (No Response)

6. Review Comments to the Author

Reviewer #1: (No Response)

7. PLOS authors have the option to publish the peer review history of their article (what does this mean?). If published, this will include your full peer review and any attached files.

Reviewer #1: No

---

## [Editor Report · Acceptance letter]

14 Jul 2022

PONE-D-22-06423R1 

Interaction-based Mendelian randomization with measured and unmeasured gene-by-covariate interactions 

Dear Dr. Spiller:

I'm pleased to inform you that your manuscript has been deemed suitable for publication in PLOS ONE. Congratulations! Your manuscript is now with our production department. 

Kind regards, 

on behalf of

Prof. Momiao Xiong 

Academic Editor

PLOS ONE